# Perceived Factors of Stress and Its Outcomes among Hotel Housekeepers in the Balearic Islands: A Qualitative Approach from a Gender Perspective

**DOI:** 10.3390/ijerph18010052

**Published:** 2020-12-23

**Authors:** Xenia Chela-Alvarez, Oana Bulilete, M. Esther García-Buades, Victoria A. Ferrer-Perez, Joan Llobera-Canaves

**Affiliations:** 1Primary Care Research Unit of Mallorca, Balearic Islands Health Services, 07003 Palma, Spain; oana.bulilete@ssib.es (O.B.); jllobera@ibsalut.caib.es (J.L.-C.); 2Department of Psychology, University of the Balearic Islands, 07122 Palma, Spain; esther.garcia@uib.es (M.E.G.-B.); victoria.ferrer@uib.es (V.A.F-P.)

**Keywords:** hotel housekeepers, stress, occupational health, job demands-resources model, qualitative research, work-life balance, gender perspective

## Abstract

Tourism is the main economic sector in the Balearic Islands (Spain) and hotel housekeepers (HHs) are a large occupational group, in which stress is becoming a major issue. This study aims at exploring in-depth factors perceived as stressors by HHs and key-informants, and their effects on work-life balance (WLB). A qualitative design with phenomenological approach was used, conducting six focus groups with 34 HHs and 10 individual interviews with key-informants. Results were analyzed adopting the job demands-resources model and a gender perspective. High demands, e.g., work overload, time pressure, physical burden…, lack of enough resources and little control, derived from role conflict, unexpected events…, were the most important factors explaining HHs’ stress. Additionally, this imbalance was perceived as leading to health problems, mainly musculoskeletal disorders. Working schedule was mentioned as a facilitator to WLB, whereas an imbalance between job demands and resources led to work-home conflict, preventing them from enjoying leisure time. Multiple roles at work and at home increased their stress. HHs experienced their job as invisible and unrecognised. Regarding practical implications, our recommendations for hotel organization include reducing workload and increasing resources, which would improve the job demands-resource balance, diminish negative mental and physical outcomes and improve WLB.

## 1. Introduction

Tourism accounted for 13.4% of employment in Spain and 25.6% in the Balearic Islands in 2019 [1]. Within the hotel sector, hotel housekeepers (HHs) are the second-largest occupational group [2]. An estimated 13,000 HHs work in the Balearic Islands [3], a female-dominated sector in Spain (almost 100%) [4]. Although working conditions in the tourism sector have been studied, the case of HHs needs special attention due to the hardness of their job. The work of HHs consists basically of cleaning and tidying up the guests’ rooms and bathrooms, and communal hotel areas. HHs are exposed to various risk factors related to work including physical [5], chemical [6], biological [7], and psychological [8] risks. Being exposed to these risks, HHs’ visits to the doctor are frequent due to their health problems—musculoskeletal, above all [9,10,11,12,13], but also due to anxiety and stress. These health problems often lead HHs to self-medication [4,14].

Since exposure to psychosocial job hazards may result in a higher risk for HHs’ health status [15,16,17,18], it is important to advance our understanding of their experience of stress. In this context, our qualitative study aims to analyse in-depth HHs’ experiences and perceptions of stress from a gender perspective. Thus, we take into account a comprehensive person-centred approach, by exploring the implications of HHs’ work and stress for their work-life balance. The results will hopefully provide evidence and guidance as to how to improve their working conditions and, consequently, to diminish their stress levels and improving their well-being.

### 1.1. Hotel Industry, Stress and Gender

Tourism is one of the most stressful work settings [19]. Research has identified factors leading to high levels of stress, burnout, and exhaustion in the hotel industry, such as work overload [17,20,21], time pressure [14,22,23], work intensification [4,24,25], lack of flexibility, conflict, low task control, the work environment [21], work/tools equipment, support from supervisor and body pain [26]. HHs’ job also implies ‘emotional requirements’, which may have positive effects (e.g., work self-efficacy) when workers like dealing with clients, or negative effects (e.g., stress and emotional exhaustion) [27]. Overall, research shows that due to high demands—both physical and cognitive- and a low decision margin, working as a HH is a risk factor for developing mental health problems [15,16], like stress. Negative effects of stress on workers’ physical and mental health have been reported, such as minor depressions, anxiety, chronic mental health problems, cardiovascular diseases, and musculoskeletal disorders [15,28,29]. Additionally, stress is one of the most important psychosocial risk factors due to the amount of time spent in the workplace and to economic resources linked to work [20].

In the hospitality sector, most of the jobs are unskilled and held mostly by migrant and young women under temporary contracts and with low wages [30,31]. This socio-demographic profile coupled with the working conditions make these workers more vulnerable to many stressors, such as violence, discriminatory attitudes and conflicting demands from clients and supervisors [30]. Certain groups are more likely to be in a disadvantageous situation, and therefore experience more stressors as well as limited access to resources to cope with them [32]. In this sense, gender and gender social conditioning is relevant. Horizontal and vertical segregation make women exposed to different hazards; for instance, feminized jobs include more repetitive tasks [33]. Also, women tend to be more sensitive to certain task demands –tight deadlines, quick work, complex tasks, etc. [34] and occupational categories with a large proportion of women tend to have lower wages and prestige, worse working conditions, and fewer options for career development (e.g., work-related stressors [33]).

Research provides mixed evidence about the relationship between gender and stress. Data from the 6th Spanish National Working Conditions Survey (2015) [35] showed that 20.5% of working women have stress, whereas 14.3% of working men do. Garrosa and Gálvez [36] showed that the same stressors might affect and be experienced differently by women and by men, whereas Gyllensten and Palmer stated that it is difficult “*to draw any firm conclusions regarding the role of gender in the level of workplace stress*” ([37], p. 284).

Other authors relate higher levels of stress for women with work-life conflict i.e., “a process whereby contextual demands in one domain drain personal resources, leaving insufficient personal resources to function optimally in the other domain” [38]; thus, multiple roles can lead to what has been called the role strain or overload hypothesis [39]. It is important to observe that gender inequalities in private life—where women are the main person in charge of domestic chores and caring for dependents—can place additional stress on working women [33]. Spanish women devoted on average four hours per day to domestic tasks, whereas men devoted 1 h 50 min [40].

### 1.2. The Job Demands-Resources Model and the Interdependencies between Work and Family Life

Several theoretical frameworks have explained the development of stress in work settings and described adverse health effects as a result of demands-resources imbalances [22,41,42,43]. The effort-reward imbalance model (ERI) posits that the lack of balance in work settings between demands (e.g., workload and work pace) and rewards (e.g., salary and support from colleagues and supervisors) results in emotional distress [10,43]. The job demands-control model (DCM) [41] predicts mental strain as a result of the interaction of job demands and decision latitude, i.e., the potential the worker has deciding how to meet job demands. More recently, the Job Demands-Resources (JD-R) model has been used to analyse different occupational settings explaining the relative contribution of specific job demands and resources to well-being, stress, burnout, and work engagement, and to assess gender differences in these outcomes [44,45].

The JD-R model classifies working conditions into job demands and job resources. Job demands refer to those “physical, social, or organizational aspects of the job that require sustained physical or mental effort and are therefore associated with certain physiological and psychological costs”; and Job resources refer to the “physical, psychological, social or organizational aspects of the job that may do any of the following: (a) be functional in achieving work goals, (b) reduce job demands at the associated physiological and psychological costs; (c) stimulate personal growth and development” [44] (p. 501). According to the JD-R model, the exposure to chronic high job demands leads to health impairment (i.e., stress and burnout), whereas the limited availability of job resources leads to undermining employees’ motivation, which in turn can result in disengagement from work and low feelings of professional efficacy [44,46].

Furthermore, the JD-R model has been applied to examine interdependencies between work and family life [47]. Women roles at home and within the family are characterized by high psychological demands and low control [48], making female workers more vulnerable to stress. Besides, Liu and Cheung [47] found that job demands were positively related to conflict between work and family life, whereas job resources were positively related to positive experiences of work-life balance (WLB). Overall, evidence provides strong empirical support for the JD-R model in different occupational settings, and for its applicability to the stress arising from the work-family interface.

Given its strengths, we used the JD-R model as the main theoretical framework for the analysis of the data; it allows identifying not just the demands which have been reported widely [14,20,26,28,49,50,51], but also the resources available to HHs [28,49,50]. Being a feminized collective, we understand that the analysis of their experience of stress has to include the interactions between work and family spheres and an approach from a gender perspective. A gender perspective allows us to observe a potential interaction between labour and domestic spheres that affect HHs’ stress. As suggested by Hobfoll et al. [52], considering the stressors that are unique to employed women is important in order to better understand the specific needs of working women. Finally, a qualitative approach allows us to identify HHs job-specific demands, resources, stress levels, and consequences.

Based on the JD-R model, the main aim of the study is to qualitatively explore HHs’ perceptions and experiences of stress and its consequences on health and work-family life from a gender perspective. The specific objectives of this study are to identify: (i) factors of the work environment perceived as stressors (demands); (ii) factors of the work environment perceived as resources; (iii) the perceived outcomes of the (im)balance between demands and resources; and (iv) how distress affects HHs’ personal and family life.

## 2. Materials and Methods

### 2.1. Design

A qualitative design with phenomenological approach was undertaken to explore HHs’ perceptions and experiences of stress. We conducted individual semi-structured interviews and focus groups (FGs). Individual interviews with HHs and a supervisor allowed us to explore the main topics related to their job; interviews with other key informants allowed us to broaden the perspective of the phenomenon and to contrast different explanations and experiences given by HHs. On the other hand, FG was the method to gather information about the experiences of HHs related to work, identify shared experiences and perceptions, and to go in depth in their perceptions and experiences of stress as well as its implications for their personal life.

### 2.2. Participants and Setting

This project was undertaken by the Department of Primary Health Care in the Balearic Islands. HHs participating in FGs were recruited through purposive sampling: GPs in different health care centres identified potential participants according to sociodemographic and labour variables, and informed them about the research. This was the most feasible and effective way to contact them and get their participation. Afterwards, researchers contacted and invited them to participate in FGs and set the date. Selection criteria included being 18 years old or older and having worked as a HH during the last season (2017). Additionally, profiles regarding different variables—age, years working as a HH, hotel star rating, and kind of contract (permanent, temporary or permanent-seasonal employment contract)-were included in order to ensure generating rich information.

Key informants were selected through purposive sampling to obtain different perspectives and rich information. A code was also assigned to them to guarantee their anonymity.

### 2.3. Data Collection

Empirical material was collected through FGs with HHs—conducted in different health care centres—and semi-structured interviews to key informants conducted between January and June 2018. Four FGs were undertaken in Mallorca, one in Menorca and one in Eivissa. FGs ranged from 60 to 90 min and interviews from 25 to 80 min; all of them were conducted by the first author. Interviews were recorded digitally and FGs were video-recorded as well. Data collection was undertaken until saturation of the information.

Based on the literature review, we developed a script to explore the areas identified as relevant, and to approach them in a similar way across all FG and interviews. The initial script was completed as the data collection was progressing. Although some areas or questions were adapted according to key-informant profiles, the areas approached in FG and interviews were as follows:(1)Characteristics and organization of work of HHs.(2)Positive and negative aspects of HHs’ job.(3)Equipment and materials available.(4)Relationships among hotel workers and among HHs.(5)Factors of stress.(6)Health problems.

### 2.4. Data Analysis

FGs and interviews were literally transcribed. An alphanumeric code was assigned to each HH in order to guarantee the anonymity but also to be able to identify the contributions of each person and each FG. Each contribution was identified by “HH” (meaning ‘hotel housekeeper’) and by two numbers separated by a dot: the first number correspond to the FG (and it ranges from 1 to 6) and the second number pertain to the individual who made the contribution.

The contents of FGs and interviews were analyzed jointly, aiming at identify both similarities and differences in the narratives. Content analysis was undertaken emphasizing the meaning of the text. First, a code tree was elaborated according to the objectives of the research, inspired by the JD-R model categories and the reading of some FGs transcriptions. This code tree was checked by a second researcher. In order to guarantee internal validity, both researchers encoded and analyzed the transcriptions separately. Finally, analysis of each code was discussed and conclusions were agreed. Software NVivo11 (QSR International, Burlington, United States) was used to assist this analysis.

### 2.5. Ethical Approval

The study was approved by the Balearic Islands Research Ethics Committee (IB3738/18 PI). An information sheet and informed consent was given to the participants before undertaking the interview or FG. Signed agreement of the forms was compulsory to participate.

## 3. Results

Six FGs and 10 interviews with key-informants were carried out. A total of 34 HHs participated in FGs—between four and eight in each FG. A total of 64 HHs were invited: 20 of them rejected participation—due to incompatibility of schedules, being abroad, etc. and 10 had agreed to participate but did finally not come to the FG.

Socio-demographic characteristics of FGs’ participants are displayed in Table 1 and key-informants profile in Table 2.

The results are presented under key thematic headings of the JD-R model: demands perceived, resources identified, outcomes reported of demands-resources imbalance, and consequences over WLB (Figure 1). Links to gender theory and stress models are identified in the discussion section.

Stress was the most commented job-related negative aspect and it was an issue arising in all FG and interviews conducted with HHs. It emerged mostly when HHs were asked to mention some characteristics of their work and a negative aspect of their job.

### 3.1. Demands Perceived

#### 3.1.1. Tasks and Work Overload

The main stressor perceived by HHs was work overload: it was related to the number of check-out rooms to clean, the state of the rooms—e.g., untidiness-, unexpected events—e.g., a flood, and the replacement of amenities and shifts.

HHs perceived they had to clean too many check-out rooms, which had to be cleaned in depth and be ready at a certain time for the new clients. Regarding the state of the rooms, the director of HR and the head of the occupational health department admitted the fact that untidiness altered the daily work load. However, from the HHs’ perspective, this was not taken into account when calculating working times.

HHi4. There are some stipulated times that we really don’t know where they come from. Because nobody has ever come to ask how much time you spend in one room. They calculate that in ten or fifteen minutes you have to finish an apartment no matter how full it is, no matter how much crap there is, whether they are clean or dirty or have kids whose toys are thrown all over the place. You have to leave all of that in a reasonable condition.

OHealth. The big problem that we find with the HHs is the diversity of all the activities. In the same hotel, two identical rooms, the workload is different, right? One thing is the family that comes back from the beach with all their things, sand, the toy boats… another thing is the family that didn’t go to the beach that day and their room is completely different.

Regarding amenities, some hotels had a mini-bar in rooms; in most cases, HHs were in charge of replacing the products consumed by clients. Most HHs perceived that amenities had increased during the last years and some of them thought that these tasks should be done by a barman.

HH5.3. Every single year they increase our workload; now we have to check the mini fridge, in every room check for coca cola, water, juices, crisps, candies, tea… everything. It’s [now all part of] our job [but it didn’t used to be].

Prev. They restock the minibar. They have a cart. They have bottles, they restock amenities, papers…Yes.

Another factor of workload perception was “rooms’ changing”, a practice that occurred when the reception desk changed the room assigned to clients when they did not like the room they had in the first place. HHs complained because this meant they had to clean a room twice. This practice deteriorated the relationship between the reception desk and housekeeping department.

HH6.8. Now since they [hotel management] don’t want to compensate them, they change rooms. (…) Instead of six check-outs, you have ten. And that’s very hard for a HH.

Some issues identified by HHs and interviewees as causes of the increase of workload during the last years were:(1)The reduction of workforce, which implied having to assume more rooms to clean; not having enough staff to replace HHs on short sick leave; not having their days off when someone was ill; carrying out tasks not of their responsibility such as refilling mini-bars, etc.

PInsp. I think I have only found one who told me they were still the same [number of workers] as always (...). There were 15 [workers], but with the crisis they went down to 10; [but] the work was still completed [even with less staff]. Next year they went down to 8; [but] the work was still completed.

OHealth. A long term leave, (…) you get a contract, or go through a temp agency, or temporary contracts (…). Now, for example, it’s 7 in the morning and the director receives the “hey, I don’t feel well” and now there’s no time to-. So now the work that person had has to be divided among all the HHs including the ones from the afternoon shift.

(2)The shortening of the length of stay of clients, which increased the number of check-out rooms to clean.

HH5.4. Since the crisis, people stay for two days, for three days, there are a lot of check-outs, they come, they go, this one has gone, this one hasn’t left yet… And it’s… besides the work, it’s stressful.

#### 3.1.2. Time Pressure and Work Pace

Most HHs in FGs and interviews perceived time pressure. The planned time to clean rooms was very tight, and neither displacements between rooms nor unexpected events were taken into account when calculating how many rooms per day HHs were able to clean.

Insufficient time was perceived as having negative consequences for their health, such as mental strain and physical exhaustion. Other consequences they reported were not being able to have lunch or use the toilet while working, not being able to use the individual protection equipment properly (e.g., gloves, glasses, mask,…), not being able to help colleagues and to do tasks as properly as they would like or were required to. Time pressure and work pace, coupled with lack of equipment –i.e., bed linen- were perceived as leading to interpersonal tensions among HHs:

HH5.1. I find it very stressful (…) the small amount of time they give you to finish such a complicated task.

HHi2. It’s not even an option for a person who’s working to be able to head down to pee because there’s no time. They don’t eat anything, and if I head down to eat, I can’t really eat because of all the stress I have.

SUP. While you go to another floor, several minutes have passed. Those minutes are dead time for the company. They don’t figure that in.

This point of view was supported by the physician from the government’s inspection service. Additionally, the head of the occupational health department accepted that sometimes HHs told him not having time for lunch. Meanwhile, the head of the occupational risk prevention service stated that the housekeeping department was not the most stressful department in the hotel.

PInsp. Some tell you that they aren’t allowed to eat. (…) Come on, if you’re there from 7.30 until 4, at some point you have to eat. “Yes, but that means more of your time at work before you can leave. And you really just want to get home”. That’s what most of them tell me.

OHealth. It’s true that they sometimes tell us “I don’t even have time to head down”. (…) It’s also true that we analyze the workload and it’s not too different from the one who does have time to head down to eat.

Prev. In the room service department where the HH works alone at her pace, she has to finish 15, 16, 17, 18, however many rooms, but at her pace. I don’t see how that can be as stressful compared to, for example, [kitchen].

#### 3.1.3. Role Conflict

HHs and the supervisor reported some situations where they received demands from different departments or people. These situations were perceived as stressful because HHs had to deal with these demands, sometimes contradictory, in a context of shortage of time, implying to move and undertake tasks not included in their original working plan. Demands received were:(1)From the reception desk: sometimes clients were sent to a room that was not ready yet, so HHs had to talk to clients without having nor enough time neither knowledge to explain the situation. Additionally, they thought that giving this kind of explanations was not their responsibility;(2)From clients, who asked for a blanket, for a pillow, for hangers, etc., so HHs had to interrupt their duties and go for them;(3)From the supervisor, who at the same time may receive demands from clients, from the reception desk and the dining hall- asking to clean a certain place.

HH6.1. The client says: “Hey, clean my room, I’m taking up my child to take a nap”. Okay. You have rooms on another floor. The director: “Hey, the clients on such and such floor want to check-out”. You head over there. You find a client in the hallway: “You haven’t cleaned my room”.

SUP. We are very, very, very stressed. Then there’s reception asking if a certain room is ready, and this client who wants a blanket, and this client who wants this, and this client who wants that. You have to go and take those things to them.

A consequence of role conflict was the feeling of being maids of other hotel workers and a lack of recognition. HHs stated that their colleagues or supervisors expected or ordered them to clean areas not of their responsibility, such as the dining hall and bar.

HH4.3. We’re maids to our own co-workers. They tell you to clean up their own work space.

SUP. In any case, management doesn’t tend to appreciate the work that HHs do. Mostly because, of course, when the girls do a good job, they never value that.

HH1.2. They don’t really appreciate us. We’re the maid… we’re the nannies for everybody. Because if the kitchen is dirty your own co-workers won’t help you.

The issue of receiving different demands was absent from the discourse of the director of HR, the head of the occupational risk prevention service, and the head of occupational health department, showing little awareness about details of the every-day work organization of HHs.

Another aspect regarding role conflict was “rigorousness”. Hotel managers and supervisors demanded HHs to do the work properly, but also HHs showed pride and satisfaction in doing their work accurately despite not having enough resources. The combination of direct supervision, the thoroughness demanded, and the lack of time was referred as very stressful.

HH1.3. From my point of view and in my experience, which is not very much, they give us a lot of work given what a good job they expect us to do.

#### 3.1.4. Physical Burden

HHs considered their job as physically demanding. It included pushing the trolley, moving furniture—in order to clean the room properly or to put furniture back in place-, moving beds, carrying bed linen, and cleaning the panes, among others. This physical burden was perceived as cause of some of their physical problems, basically musculoskeletal disorders (see 3.3 Outcomes Reported of Demands-Resources Imbalanced). Physical burden was also recognised also by key-informants interviewed:

HHi4. And we use duvets. (…) imagine in every apartment having to change the covers for five or six duvets. By yourself.

OHealth. We know that the work done by the HHs is physical work that requires effort.

Sometimes, hotel facilities, furniture and work materials and equipment could increase the physical burden (e.g., steps that prevented the use of the trolley, furniture too heavy to be easily moved, too big or too small trolleys, etc.).

HH2.2. I had a better cart and they exchanged it for a much smaller one, but nothing fits in it.

HRDir: Hotels that have carpet, it’s much harder to move the bed. While hotels that don’t have carpet are better, or where the beds have rollers.

#### 3.1.5. Personal Characteristics

Age and physical status were perceived by HHs as factors that prevented them from doing their work properly, to face the workload effectively and to keep up with the work. Moreover, HHs perceived their educational level as a barrier to get another job: they mentioned they could not have many job or career opportunities because many had completed compulsory education only. The combination of these factors made HHs have different expectations regarding their future. The youngest HHs did not see themselves working as HH all their life; the HHs in their 40’s and 50’s were wondering whether they would be able to work until the retirement age; and the oldest HHs were just waiting for their retirement.

HH5.4. Psychologically, I’m okay to work. But physically I hurt.

In summary, demands perceived by HHs as stressful were related mainly to work overload, work organization (time pressure, work pace, and role conflict), and the physical burden.

### 3.2. Resources Identified

#### 3.2.1. Salary

Most HHs agreed that their wage was acceptable compared to other jobs, but it was not enough in relation to their workload. Thus in economic terms they thought they were not fairly rewarded.

HHi3. Financially, we could say a HH earns a decent salary.

HH3.5. The salary is not what you’d think. For the work we do, it’s badly paid. (…) We’re the bottom rung of the ladder. Earning less than a technician, earning less than a reception assistant…

#### 3.2.2. Training/Promotion

When talking about training, most participants’ comments referred just to courses on prevention of occupational hazards. HHs highlighted that they already had this kind of knowledge—so, these courses were not necessary anymore- and complained that attending training did not imply a reduction in workload, and thus, involving an increase in time pressure and, consequently, in their stress. Besides, HHs reported that other types of training were not useful or did not exist at all and recognized few possibilities for promotion or improving their skills. This point of view was supported by some key-informants, such as the director of HR and the head of occupational risk prevention service. Thus, training, which is usually a resource, is lacking for this occupational group.

HH6.5. I’m stuck in a rut because there aren’t really many opportunities for advancement or to level up or get a promotion. (…) There is no professional development at all.

Prev. All departments have their general training for occupational risk prevention and specific training for the job. Minimal, once a year each one. (…) We try to modify it every year or every two years, because the thing in job training (…) is that you are always explaining the same.

#### 3.2.3. Control

Control over work or autonomy is considered a key resource in the JD-R model and aspects related to it were very mentioned in the discourse of HHs. The most important factors reported as facilitators of control over work and autonomy were as follows:(1)Possibility of choosing the order in which to clean the rooms.(2)Knowing the habits of clients (e.g., time they got out of the room).(3)Days having less work, they could advance other tasks. The supervisor interviewed supported this practice as well.(4)Having certain rooms assigned to the same cleaner everyday allows them to have more control over the workload throughout the days. The HR director interviewed supported this view:

HH4.7. What’s more, you get organized yourself. What I mean is if a client tells me that they don’t want me to clean their room that day and I have a bit more flexibility with my time, I’ll do a little extra in the room I’m cleaning, for tomorrow. To get ahead.

HRDir. Because each one likes to have their space and to have it under control.

In contrast, some factors already mentioned, such as the role conflict arising from receiving different demands from different departments or clients and the state of the rooms were commented as making their control over work more difficult:

HH3.1. Go back to each of the do-not-disturbs, go back to all the ones who told you they were sleeping, go back to all the ones who said “I can’t right now”, “come back later, I’m eating”, “ten minutes, I’m heading down to the beach”. And that puts me behind schedule.

HH5.3. Because you really don’t know what you’re going to find when you open the room, if you’re going to be there for fifteen minutes, twenty or thirty.

#### 3.2.4. Satisfaction and Recognition

HHs stated that schedule was the most positive feature of their job: it was continuous—7 a.m. to 4 p.m. approx.- and it allowed them to better balance work, family and personal life:

HHi2. For me the schedule is the best part of the job because you have afternoon off, even though you’re really tired.

Clients were a main source of satisfaction and recognition of HHs’ work. Some clients leave a tip, others tidy up the room to make it easier to clean it, some congratulate them for the work done, etc.

HH5.2. It is gratifying because the client knows you and says: “Oh, very nice, very nice, we like you.” And they’re pleased to tell you, you know? And then you feel good about the work you’ve done.

Nonetheless, as aforementioned, interactions with clients were not always satisfactory: in a context of shortage of time, these interactions were demands and usually experienced as stressful:

HH4.4: When new clients check-in and the card doesn’t work and they cannot open the door and you are like “Oh, don’t look at me, don’t see me.” And you see the old people: “Hey girl, this does not work.” And you are like “Oh, don’t call me.”

The feelings of satisfaction stemmed also from the importance and meaning HHs gave to their job and the pride in doing the work accurately. Other positive aspects were relationships with clients and the wage, as aforementioned.

#### 3.2.5. Social Support

Social support includes the relative presence or absence of psychological support resources from other significant people. From narratives from FG and interviews, it can be stated that HHs considered social support important: above all, HHs mentioned instrumental and emotional support.

Regarding instrumental support, helping among HHs was mentioned as a quite unusual and informal practice, not explicitly proposed by hotel management. HHs reported that workload prevented them from helping their colleagues. Nevertheless, in most cases they tried to help colleagues when needed—sometimes without informing nor the supervisor neither managers, and some HHs pointed out they would like to work in pairs because some tasks would be easier to perform; so, most HHs would support that working in pairs was a more common practice. The head of occupational risk prevention service mentioned that hotel management hired more HHs when work load was unusually high. However, these “extra HHs” sometimes were not seen by HHs as helpful because they were not permanent employees and did not know how to do the work properly. Finally, HHs reported that in some cases the supervisor decided who was to be helped when a HH finished her work ahead of schedule:

HH1.1. We can’t help each other any longer because it’s impossible. Because she’s just as busy as me.

SUP. If they see that one of them isn’t doing well because she’s had problems because check-outs went badly, they lend each other a hand.

Regarding emotional support, some HHs mentioned that relationship with colleagues was one of the most positive aspects of their job. However, sometimes there were interpersonal tensions due to the high pace work and time pressure:

HH4.2. The relationships tend to be good.

HH4.4. You fight, argue, laugh, kiss

HH4.7. But the stress and the exhaustion

HH4.4. Why? Because they took your mop away, the mop is mine, the mop is yours, okay fine, you take it.

HH4.7. What brings that on? Stress. Exhaustion.

#### 3.2.6. Experience

Experience working as HHs was the personal resource that helped them to cope with the workload. However, as they were getting more experience they were getting older as well. Thus, age was also perceived as a factor preventing them from doing their job properly:

HH4.6. You have many years of experience, you know how to do your job well, but you don’t have the time or the energy to get it done and it produces anxiety where you just can’t take it anymore. You finish and leave work in tears.

Summarizing, the most important resources related to work identified by HHs were salary (when compared to similar jobs), recognition from clients, satisfaction with the working schedule, satisfaction with a well-done work, and their experience working as HH. Other resources such as helping among HHs, social support among HHs, and training are important resources, but currently present to a lesser extent in their jobs.

### 3.3. Outcomes Reported of Demands-Resources Imbalanced

Most HHs reported different mental and physical outcomes related to their work. First of all, most HHs reported being nervous or stressed the whole working day because of work overload. This perception of not being able to cope with workload made them feel sad, not wanting to go to work, and psychologically burned out. Another source of stress was having to deal with clients sent to the room before it was ready. Consequently, most HHs described their job as making them “anxious”.

A second mental outcome reported was being unable to disconnect from work due to work overload and the anxiety described above. This implied basically difficulties to sleep well and to enjoy leisure time.

Third, HHs reported tension among co-workers and a bad work environment. From their point of view, not having enough time to face the workload involved getting angry with co-workers when they needed help, materials, etc.

Last but not least, HHs described many physical health problems, mainly musculoskeletal disorders. HHs perceived that these were directly related to the organization of work—e.g., a high working pace, a high workload, and physical burden. One of the most commented strategies towards health problems was self-medication. HHs explained that they were taking anti-inflammatories and analgesics and, to a lesser extent, tranquilizers and anxiolytics. This information was confirmed by health workers and the head of the occupational risk prevention service interviewed:

HH4.7. Constant stress. From the moment they give you the paper and you go home thinking about what you have to do the next day. (…) you don’t disconnect: “tomorrow I have so many check-outs, will I be able to get them done?”

HHi4. Work overload is taking a toll on our health.

HH5.2. I self-medicate (…) with pills, waiting until it really hurts so that I don’t have too much medication in my system. I’m tired of taking medication.

Prev. Young workers still don’t take anything. But yes, somebody in their late forties, early fifties who’s been working 20 odd years in the hotel industry has to take some kind of medication.

### 3.4. Work—Life Balance

A short working day—from 7 am to 4 pm with 15 min for breakfast and 30 min for lunch-was mentioned as the most positive aspect of their job because it facilitated their WLB. Actually, the working schedule was one of the reasons given by HHs to take this job. It allowed them to take care of the kids—if they had children—without leaving their jobs, to do housework, and to enjoy social life:

HH2.4. You have time for yourself, if you have kids you have time for home, for everything. To have a bit of a social life. The people who have a split shift are there almost 24 h.

HH3.1. And I started because of the schedule. I had young kids and it worked well for me not to have a split shift. That was the only reason.

Negative consequences on WLB reported by HHs were not having enough energy nor to face family demands neither to enjoy their family. Moreover, as aforementioned, HHs reported difficulties for disconnecting from work and resting when being at home, which had negative implications for enjoying personal time. These consequences on WLB were perceived as being caused by work overload and the physical burden of their job. Some reported the hardness of working as HH, being responsible of housework and having children. When arriving home exhausted from work, HHs had to go to the park with their children, cook dinner, tidy up the house, etc.:

HH6.5. I have an eleven year old son, a one year old grandson, a thirty-two year old daughter and it’s stressful because I get home and my daughter tells me she’s coming over for lunch with my grandson. I should be happy but I’m not really. And I’m like, damn, now she’s coming over and I’m screwed. Because now I have to prepare food for four, clear the table…Because she’s coming to spend time with her mother, but her mother is thinking about all that she has to do…

HHi2. Because for the person who carries such a heavy workload on the outside and inside, it’s hard to have relationships and you can’t do everything with your daughter that you’d like to.

The two days-off HHs had per week also had consequences on WLB. Different situations regarding the days-off were identified. Some HHs reported that during the summer they could take just one day off per week due to workload and lack of staff. Others stated that they could enjoy always the same two weekdays-off, while others could not and the days-off changed from one week to another, sometimes unpredictably. The two days-off were perceived as a tool to balance work, family and personal life. When these days were constantly changing, difficulties in WLB increased:

HH5.1. Because I went to the director and said: “I want to at least know one day that I’m getting off because if I have to see the doctor, they’re my personal things”.

To sum up, the working schedule is perceived as the most positive aspect of HHs’ job for WLB; however, other work factors affect WLB negatively these women. HHs felt exhausted after their working day and had problems to disconnect from work when being at home; consequently HHs did not feel as having enough strength to face family and domestic demands or to enjoy leisure time. Moreover, despite days-off are perceived as a tool for WLB, in some cases hotel management constantly changed them and make WLB more difficult.

The summary of the main findings—demands, resources and outcomes-are displayed in Figure 2.

## 4. Discussion

Previous research has focused mainly on demands of the HHs occupation [14,20,26]. Despite the high physical demands of their job, HHs report stress as one of the most relevant perceived problems related to it. In this project, we qualitatively explored in-depth the experience of HHs in their job under the framework of the JD-R model and from a gender perspective. Qualitative methods provide a useful approach, permitting understanding and “explaining social phenomena «from inside»” [53] (p.12), because participants can express themselves openly. They allow going beyond pre-stablished and standardized categories as well as capturing information about unexpected organization-specific job demands and job resources, as suggested by Bakker and Demerouti [54], and understand their levels of stress. The gender perspective is necessary given that HH is a female-dominated occupation, affected by gender roles both at work and at home. Gender roles imply that women are the main person in charge of domestic chores and have a greater total workload (domestic workload plus paid workload). Additionally, women are exposed to different hazards at work due to the horizontal and vertical segregation of the labour market, that lead women and men to hold different jobs and to perform different tasks [33]. Applying the JD-R model to analyse stress experiences reported by HHs and key-informants proved useful, as the model offers clear and wide enough categories to codify different elements, and allows for the inclusion of both work-related and domestic stressors.

Our main findings (see Figure 2) showed that high demands—e.g., workload overload, time pressure, physical burden, etc., lack of resources in general and little control—derived from role conflict, unexpected events, the state of the rooms, etc.—were the main features of the HH’s job and the most important factors explaining HHs mental strain and physical health problems. Imbalance between demands and resources—and the stress caused by it is maintained for months and year after year, what may lead to health impairment process and to physical problems [17,28], as perceived by HHs in our study. Further, stress negatively affects HH’s personal and family life.

Work demands. Our results pointed out that work overload, time pressure and work pace, role conflict and the physical burden were perceived by HHs as the main stressors. Workload [16,20] and interpersonal tensions [17] were two of the most frequent stressors experienced by HHs and other hotel employees. Besides their daily quota of rooms to clean, usually within a stipulated timeframe, HHs take care of unexpected tasks, such as last minute situations, cleaning late check-outs [16], or replacement of workmates [55]. Moreover, the design and decorations affect HHs’ workload [56] as well as the physical burden of the job—i.e., having to move heavy furniture-. Hotel facilities and rooms are designed for the client to enjoy, rather than to make cleaning tasks easier [26,55]. Although workload was not a significant predictor of HHs’ stress in the study of Hsieh [26], Cañada reported [4], according to our results, the perception of HHs of the increase in workload during last years, mostly due to workforce reduction.

Time pressure and a high work pace were in part caused by the fact that the housekeeping department receives additional pressure by other departments in order to offer quick and high-quality services [15]. These employment conditions lead to time pressure, a central factor explaining stress identified in previous studies. HHs interviewed in Hsieh et al. [14] constantly experienced time pressure at work; in other studies it was also identified as a relevant stressor [4,50], as a negative aspect of their job [51,57], and as a concern [49]. Moreover, García-Herrero et al. [34] reported that women were more sensitive to quick work; in this sense, some HHs in our study expressed this time pressure as having to work quickly throughout the whole working day.

Regarding role conflict, it stemmed from receiving requests—sometimes conflicting—from different hotel departments, from colleagues and from clients. Perceptions of occupying an unskilled and ‘servile’ job, performing women’s work and being the lowest rung among hotel workers [20,49,50,51,57,58] were, in the eyes of HHs, provoking additional requests by workers of other departments who asked HHs for doing tasks not of their responsibility. The results pointed out that HHs perceived that their colleagues did not take into account their workload and did not value the work they did, which in part contributed to HHs’ feeling of ‘being maids’, similar to the feelings of exploitation reported by Kensbock et al. [49]. For all these reasons, hotel housekeeping job is experienced as an invisible an unrecognised job, comparable to domestic work; additionally meanwhile discrimination and stereotyping are stressors identified in literature as being more common in women [37].

Moreover, HHs’ interactions with clients during their workday can become an added pressure [59]—e.g., having to manage their feelings or hide them [23], and having to keep a kind and happy attitude [25]. According to our findings, not only keeping a kind attitude is a source of stress, but also the feeling of losing their time in a context of shortage of time.

Our findings suggest that interpersonal tensions were perceived as a result of stress more than a cause of it: they were experienced by HHs as derived from circumstances such as lack of equipment, time pressure and role conflict.

Regarding the impact of demands on their physical status, HHs perceived that high work demands led them to fatigue, relation supported by evidence: similar patterns have been observed in different occupational settings—nurses, teachers, manufacturing and chemical industry, air-traffic controllers, in which unfavourable job demands were positively related to exhaustion [44,46,60].

Resources. On the positive side, main resources reported by HHs were recognition of clients, social support among HHs, job experience as HH, some control over the work, intrinsic rewards, and, to some extent, salary. HHs reported that clients were the main source of recognition and it was described as very important by HHs in FGs, generating self-pride and satisfaction as reported in Nimri et al. [61]. Nevertheless, our results pointed out that emotional labour involved in interactions with clients had both positive and negative outcomes, as it might provoke stressful situations due to clients’ demands.

Besides, social support from colleagues was mentioned as a positive feature of HHs’ job too. Although helping among HHs was not common due to workload, relationships among them were considered one of the most positive aspects. In this sense, Hunter Powell and Watson [50] identified that 83% of the HHs valued interpersonal interactions, which in turn have been recognized to be more important for female than male [62].

Working experience as HH was valued as a personal resource, because it implied know-how to do the tasks properly and quicker; this positive attitude was expressed in Nimri et al. as “challenging due to the physicality and the monotony of the room attendants’ daily tasks” [61]. On the negative side, HHs highlighted that getting older implied a worse physical state and having less energy to cope with work demands.

HHs valued factors that allowed them to perceive a certain control over the work—e.g., choosing the order to clean the rooms, having certain rooms assigned, etc. These resources raised as strategies in order to counteract to some extent the lack of other important resources in their work setting—such as time.

Although salary is considered as a resource, it has been reported as a negative factor in HHs’ work. Yilmaz [63] and Kensbock et al. [49] found that HHs were not satisfied with their wages, nor did they feel that that salary compensated their efforts. In this line, Chiang and Liu [16] reported that HHs wanted to receive some extra-pays, action that would reduce their job stress. Although HHs interviewed in our study emphasized the negative aspect of the salary, not being enough for the workload they supported, they also commented a positive one: the salary was good enough compared to other poorly qualified jobs. This may be explained by the fact that HHs in the Balearic Islands are hired directly by the hotel company, and are not outsourced, whereas in other regions of Spain and other countries—such as United States—outsourcing is a common practice and HHs have an even more precarious employment status [3,4]. Being hired directly by the company implies better employment terms, such as more stable job contracts and a living wage.

On the other hand, despite the little extrinsic reward HHs received, lack of recognition by bosses and co-workers and an insufficient salary, they felt proud of their job and found it meaningful, an intrinsic reward, as reported by other studies with cleaning staff [64,65]. This kind of jobs, labelled as unskilled and in which tasks developed are associated with the private sphere, has negative connotations socially. But the fact that in some cases these jobs are a source of satisfaction and pride for women remind us that there could be a male bias [66] in the way the value of jobs is socially deemed.

Related to the little recognition and value given to HHs’ work, HHs mentioned insufficient training other than safety instruction, as reported in other studies with hotel workers and HHs [4,20,49]. An explanation of these aspects can be associated with the fact that tasks performed by HHs are linked to the private sphere, where women carry out most of the tasks. Accordingly, these abilities and knowledge are seen as ‘inherent’ to women and training is seen as unnecessary. As a result of this lack of training, HHs highlight a lack of career progress, a stressor which is described in literature as being more common in women than in men [37].

Despite resources commented by HHs, experiences and perceptions about salary, unrecognition from co-workers and unsatisfactory interactions with them, unclear promotion aspects and insufficient training could be factors that reinforced HHs’ feeling of ‘being maids’, being undervalued and being at the lowest rung of the organization.

Regarding WLB, various HHs in our study explained that they started working as HH when they had children, because a continued working schedule allowed them to better balance work, family and personal life. However, changes in days-off made WLB more complicated, a situation described also in Cañada [4]. Working schedule advantages have been reported as being a factor more important to women than to men [67,68]. The fact that women are the main person in charge for domestic chores and children care may explain these values and preferences.

Additionally, analysis of HHs’ discourses shows that characteristics of their work and the stress suffered in the workplace have implications for the perceptions of WLB. One of the most immediate and reported effect was the fact that the working conditions leave them with insufficient resources to face domestic and care tasks, as reported in Cañada [4]. Besides, they seemed to receive very little help from partners in the domestic setting. Sexual division of labour, which implies women being in charge of most domestic work, makes it especially difficult to achieve WLB. As a result, work-home conflict—described as the interference of work life in family life [69]—emerges as one of their basic problems and has been described as leading to a lack of personal time, greater difficulties to achieve WLB, a greater level of work dissatisfaction, and higher stress levels [34,70,71].

The interaction between work and family roles has adopted two perspectives: (1.) multiple roles—worker, spouse, mother—may give meaning to one’s life and have positive health outcomes (accumulation or enhancement hypothesis); (2.) multiple roles may lead to negative outcomes on health because they are a source of stress (overload hypothesis) [39]. Findings of our study pointed out that HHs’ multiple roles have negative outcomes in HHs, by increasing their stress and emphasizing their physical exhaustion. Consistent with previous findings [4,72,73], HHs reported to be so tired after their working day that they were not able to rest when being at home, to cope with domestic chores and care responsibilities or to enjoy family life. The quality of the job and working conditions, as well as the quality of the family roles are important to determine whether having multiple roles imply positive or negative outcomes [48]. Further, multiple roles are a stressor reported more in women than in men [37].

In this sense, HHs expressed work-family conflict when admitting difficulties to cope with housework due to their working role. According to our results, research showed that work-family conflict is negatively related to well-being [74] and, in hotel employees, positively associated with job stress [71]. In addition, Matud found that, compared to men, women reported more frequently stressful events related to family and those with children experienced more psychological distress [75].

Further, the social image of the home as a sanctuary, as a place to rest, to recuperate from problems at work, a place where solidarity reigns, etc., has been challenged by feminist studies [76,77]. HHs’ narratives supported this view: home was a place where they had to continue working, cleaning, cooking, caring for others, etc., and, consequently, another source of stress.

### 4.1. Practical Implications

Our research has practical implications both for policy makers and for Human Resources departments in hotels, who may improve organization of work and working conditions. Reducing work overload and psychosocial risk factors to which HHs are exposed would diminish stress levels and work-home conflict and improve HHs’ health status. Following the JD-R model we identify some basic proposals to improve HHs’ work. First, demands and the associated workload should be decreased (e.g., reduce number of rooms, number of tasks, etc). Second, resources should be increased (e.g., plan for unexpected events in the daily working schedule; allow working in pairs or helping among HHs; improve the design and adapt furniture and hotel facilities to cleaning tasks; and hire more housekeeping staff). Third, the participation of HHs in decision-making processes related to the design of facilities, furniture acquisition, working materials, etc. would make HHs feel important in the hotel structure and may enhance their satisfaction with work and feelings of recognition. This kind of measures would increase their control over work and would diminish HHs’ perception of stress levels, improve their well-being, and probably their satisfaction and engagement with work. Additionally, practical implications regarding the reduction of the conflict between work and family life are directly related to the implementation of WLB policies and practices such as planning days-off in advance. Overall, reducing demands and increasing resources would have a positive effect on HHs’ well-being.

### 4.2. Limitations and Future Research

Recruitment of participants in FGs was through general practitioners (GPs). This could introduce a bias related to an over-representation of discourses of HHs with more health problems—those attending the health care centre more frequently—and of those more occupationally stable—GPs would not know HHs who just work and live in the area during the season. We also acknowledge difficulties for recruiting young HHs and HHs with a temporary contract. Thus, we were not able to collect their discourse in a direct way. However, this recruitment method through GPs—without any link to enterprises or unions—gave HHs confidence to report their experiences related to health and work. Further, we believe that our results reflect reality of most HHs, as HHs’ associations (e.g., Kellys) claim that stress and certain health problems—musculoskeletal disorders above all—are generalized in this occupational group.

We should note limitations inherent to qualitative studies, such as weakness of external validity of the results and limitations to measure the incidence of stress among HHs. On the other side, qualitative methods allow to go in depth in explanations of the phenomena and getting rich information and shared accounts of facts and experiences. Future research on this area should follow survey-based studies to broaden the results. In fact, a later survey was carried out in a representative sample of HHs living and working in the Balearic Islands and will contribute with quantitative data about HHs’ working conditions, quality of life and levels of stress among others. This qualitative study was useful to design data collection survey and quantitative data will allow us to contrast whether discourses reported in this study are representative among HHs.

Also, future research should consider the characteristics of specific work settings and family burden in order to explore how they influence the experience of stress and in their well-being. Different work settings, influenced by the size of hotel, leadership styles, the kind of organization of work, human resources policies of the company, etc. and different family situations (e.g., support from spouse, distribution of domestic tasks, being autochthonous or migrant, domestic burden, etc.) could lead to different experiences of stress. Moreover, and taking into account gender roles, it would be interesting to explore in-depth possible differences between men and women in perceived work demands and resources, family demands and the interaction between work and personal/family spheres.

## 5. Conclusions

HHs are an occupational group with working conditions known to be physically hard. From HHs’ narratives obtained in our research, we can conclude that their job is also psychologically demanding and very stressful. HHs perceived their job as highly demanding in terms of work overload above all, without enough resources to cope with these demands, and having little extrinsic recognition. The outcomes of these features are not only high levels of mental strain and stress but also physical problems. Moreover, the analysis of this workforce from a gender perspective shows that hotel housekeeping work is an invisible and unrecognised job, and therefore is only visible when it is not done, characteristics shared with domestic work. The analysis also shows that work-home conflict is a relevant additional factor increasing the level of stress of HHs and emphasizing their physical exhaustion; negative outcomes of HHs’ job on WLB are reported, such as not being able to enjoy leisure time or family life. Some improvements should be made in HHs’ working conditions, e.g., reduce workloads, increase control over work, increase staff, etc., in order to diminish negative mental and physical outcomes and improve WLB.

## Figures and Tables

**Figure 1 ijerph-18-00052-f001:**
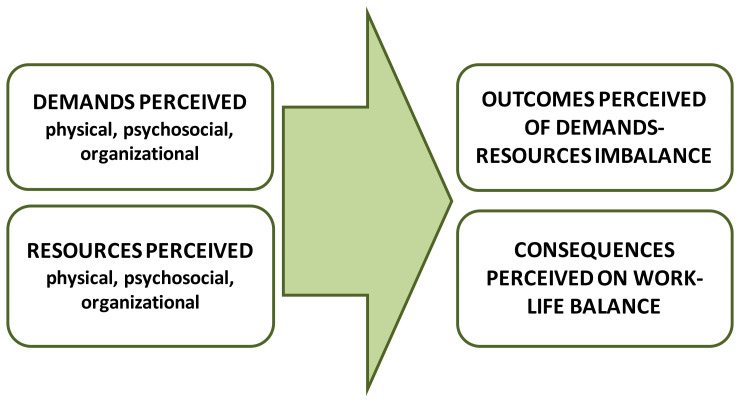
Categories of analysis.

**Figure 2 ijerph-18-00052-f002:**
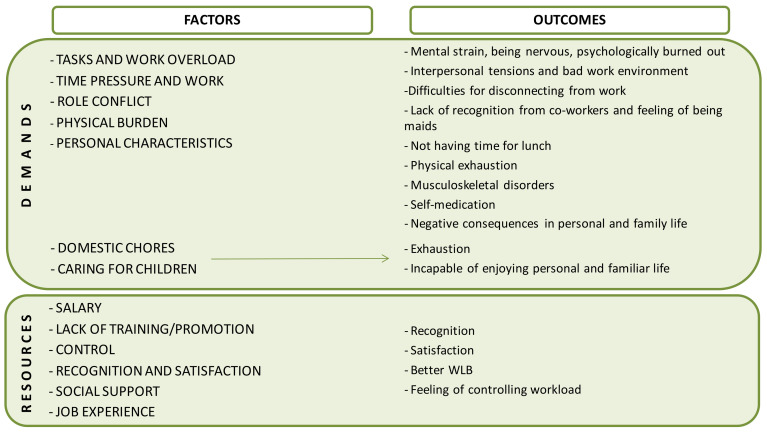
Demands and resources perceived and their outcomes.

**Table 1 ijerph-18-00052-t001:** Socio-demographic characteristics of participants in Focus Groups.

Age (*n* = 34)	<30	30 to 39	40 to 49	50–59	>59
($\bar{x}$ = 50 years old, SD = 10)	5.9%	5.9%	23.5%	52.9%	11.8%
Years working as HH	<10		10 to 14	15 to 24	≥25
(*n* = 34) ($\bar{x}$ = 19.47, SD = 11.5)	17.6%		20.6%	32.4%	29.4%
Tenure (*n* = 34)	Permanent ^1^		Recurring seasonal contract ^2^		Temporary ^3^
	2.9%		88.2%		8.8%
Hotel category (*n* = 33)	2 *		3 *	4 *	5 *
	6.1%		45.5%	42.4%	6.1%

^1^ HH work the whole year. ^2^ HH work only some months a year (usually, spring, summer and autumn), but the company commits to hiring them again the following year. ^3^ HH have a contract that last a pre-stablished number of months. * In Spain, hotels are classified according to their quality and rated by stars, from 1 (low quality) to 5 (high quality).

**Table 2 ijerph-18-00052-t002:** Codes and key informant profiles interviewees.

Code	Key-Informant Profile
HHi1	HH belonging to a union
HHi2	HH belonging to a union
HHi3	HH belonging to a HH’s association
HHi4	HH belonging to a HH’s association
SUP	HH’s supervisor
GP	General practitioner in a coastal practice
PInsp	Physician in governmental inspection service
HRDir	Director of human resources (HR) in a hotel chain
Prev	Head of the occupational risk prevention service in a hotel chain
OHealth	Head of occupational health department in a hotel chain

## Data Availability

The data presented in this study are available upon reasonable request from the corresponding author. The data are not publicly available given the confidential nature of the data, the transcripts cannot be made available in open access. because a completely anonimyzing of participants was not possible and they could be related to their job.

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
