# Peer review of "Perceived Factors of Stress and Its Outcomes among Hotel Housekeepers in the Balearic Islands: A Qualitative Approach from a Gender Perspective"

_ijerph, 2020, doi:10.3390/ijerph18010052_

Round 1
Reviewer 1 Report
My comments have been accounted for.
Author Response
We are glad that Reviewers 1,2,3 considered the paper has improved and we have covered the raised issues.
Reviewer 2 Report
The authors have correctly addressed most of the high points in the first revision of the manuscript.
Author Response

(The authors gave the same response as above.)

Reviewer 3 Report
After thoroughly reviewing this new version, I think the authors have made an excellent and meticulous effort to improve the weaknesses found in the original version. Although the sample size and the use of qualitative techniques represent an important limitation (as the authors acknowledge in the manuscript), the rigor and depth with which the authors have approached the study has led me to reconsider my initial opinion. In this sense, I believe that with the improvements introduced, the manuscript may constitute a valuable contribution. Therefore, in my opinion the work can be published in its current version.
Author Response

(The authors gave the same response as above.)

Reviewer 4 Report
This study investigates the causes of the stress of hotel housekeepers through many FGIs.
I am not sure what additional contributions this research has made over the previous research.
The listed results seem too obvious. Looking at Figure 2, where the main results are summarized, it seems likely that most people have common sense that hotel housekeepers have the demands and resources as suggested.
Since the results of this study are not unusual, the implications on page 17 do not seem to have any special suggestions.
In addition, the term 'gender perspective' is repeatedly used in this study, but it is difficult to know what gender perspective exists except that this study targets hotel housekeepers with a high proportion of women.
Author Response
Please see the attachment.

This manuscript is a resubmission of an earlier submission. The following is a list of the peer review reports and author responses from that submission.
Round 1
Reviewer 1 Report
This papers studies the problem of hotel housekeepers. Overall the paper is well written and focus on the problems of underrepresented group of people in the academic research. However, I have following comments.
- Despite the fact that a large proportion of the Island population works in hotels, the sample size of the study seems small.
- Results might be biased due to control variables, especiallly the education level of workers might help them to manage their work and stress in a better way.
- Authors may derive implications according to post Covid-19 structure of tourism.
Reviewer 2 Report
The authors have carried out a research study on the perceived factors of stress in hotel housekeepers (HHs) in the Balearic Islands. The objective of this qualitative study was to identify through a sample of HHs and key-informants: which factors of the work environment are perceived as stressors, what are the outcomes of stress, and how distress affects HHs ’personal and family life.
The major comments to the manuscript are the following:
- Although the authors correctly specify the objectives of the study, the research lacks a strong implication for clinical practice. The conclusions reached by the study were already known from previous studies.
- The study design is an appropriate, qualitative study with a phenomenological approach, carried out through the analysis of focus groups and key informants. However, the sample was obtained from the health services, through general practitioners, which entails a severe selection bias since it assumes that the selected housekeepers already had physical health problems (musculoskeletal disorders) or mental health (disorders such as anxiety, stress, burnout).
- References are not current. In the Introduction, more than 65% of the references used are more than five years old; and in the Discussion, only 2 of them are in the last five years.
The minor comments to the manuscript are the following:
- Keywords: “Hotel housekeepers” and “ job demands-resources model” are not MeSH terms. Please, check the keywords because they must be Medical Subject Heading terms.
- Introduction. The Introduction is excessively long.
- Material and Methods. In addition to the recruitment of HHs through general practitioners, I consider that the data collection should have included more predictive variables that could be associated with work stress, such as the number of rooms to be cleaned per day, work performed alone or with a partner, training, educational level, health history. Table 1 (line 200) should be in Result section.
- RESULTS. The contents extracted from the focus groups and the interviews with key informants have been jointly analyzed. The nature of the participants in each analysis (FG and Interviews) are different, with different interests and perceptions. The authors should have presented the results separately and then performed a comparative analysis (similarities and differences). This would have given more depth, beauty and strength to the results.
Explain the coding of the workers involved in the results: for example "HH5.3" what does it mean?
The subsection 3.1.2 is repeated, please check the order of numbering.
3.2.3. Social Support. It only approaches it from “helping each other among the HHs”, but social support includes the relative presence or absence of psychological support resources from other significant people, emotional, instrumental, informative, evaluative support ...
- REFERENCES. As already stated in major comments, the references are not current. Authors should endeavour to submit references from the last five years. Besides, reference number 19 is misquoted or incorrect, and And why are the original or review articles published in journals not cited as such but as online articles?
Reviewer 3 Report
The manuscript analyzes, from an approach based on the JD-R model, the main perceived stressors and their consequences in a sample of hotel housekeepers from the Balearic Islands (Spain). The study design is qualitative in nature (with in-depth interviews and focus groups). Also, it is intended to carry out an analysis adopting a gender perspective.
In my opinion, the justification regarding the relevance of the study is adequate. However, I consider that the study presents some very significant weaknesses, mainly linked to the criteria adopted for the selection of the sample. Although the authors recognize some of these aspects as a limitation, in my view they are relevant enough to prevent the generalization of the results (i.e., there are no solid evidential guarantees that the cases chosen contain the maximum expression of the phenomenon studied). It is also necessary to take into account the important limitations inherent in studies of a qualitative nature.
Likewise, the interest in analyzing the phenomenon of stress in the hotel housekeepers from a gender perspective, although (as I have already mentioned) it could be pertinent, is subject to important biases. In this sense, the researchers seem to start from the assumption that the women who develop this occupation are in a discriminated position and, consequently, are more vulnerable to stress. Although this may be the case, it is also plausible to consider that the perceived stress related to this occupation is not a product of gender differences, but attributable to the specific characteristics of the tasks performed. From this consideration, establishing conclusions based on the gender perspective is risky based on the data collected. Likewise, the conclusions obtained are based exclusively on the opinions of female participants.
These important weaknesses lead me to discourage the publication of the study. In addition, other aspects of the study of lesser consideration are indicated, but which would also require review:
Introduction
- The main job resources (physical, psychological, social, and organizational) identified by the previous investigation must be specified.
- There should be more consistency between study objectives and analysis categories. The latter are based on the JD-R model. However, it does not appear that the objectives take this model as a reference (eg, resources constitute a category of analysis. If this is the case, it should also constitute an objective.
- Decimals should be indicated with points, not commas.
Material and methods
- Did the members of the FGs know each other previously?
- The sociodemographic characteristics of key informants should be specified.
- The characteristics of the semi-structured interview and the FG should be explained in greater depth: topics addressed, questions asked, etc. (i.e., discursive axes).
Results
- Demands perceived: It would be more clarifying if the analysis of this factor were organized according to the following categories: physical, social, and organizational.
- Resources identified: organize by categories: physical, psychological, organizational, and social.
Discussion
- The interpretation of the results based on the JD-R model must be deepened.
- Figure 2 should be included in the results section.
- Practical implications: some measure should be considered to reduce the conflict between work and family life.
- Limitations regarding qualitative methodology should be noted.
Reviewer 4 Report
This paper tries to find factors affecting the stress of hotel housekeepers using the job demands-resources model.
However, I can not recommend the publication of this study. The results seem to be too obvious. And I think some survey-based statistical analyses should be followed.
In order for this paper to be published, the authors should let the readers know the importance, necessity, and contribution of this study.